# Luminescent Papers with Asymmetric Complexes of Eu(III) and Tb(III) in Polymeric Matrices and Suggested Combinations for Color Tuning

**DOI:** 10.3390/molecules28166164

**Published:** 2023-08-21

**Authors:** Roberto J. Aguado, Beatriz O. Gomes, Luisa Durães, Artur J. M. Valente

**Affiliations:** 1LEPAMAP-PRODIS Research Group, University of Girona, Maria Aurèlia Capmany 61, 17003 Girona, Spain; roberto.aguado@udg.edu; 2University of Coimbra, CQC, Department of Chemistry, Rua Larga, 3004-535 Coimbra, Portugal; gomesbeatriz642@gmail.com; 3University of Coimbra, CIEPQPF, Department of Chemical Engineering, Rua Sílvio Lima, 3030-790 Coimbra, Portugal; luisa@eq.uc.pt

**Keywords:** cellulose, heavy metals, lanthanide ions, luminescence, o-phenanthroline, paper analytical devices, poly(acrylic acid), polyfluorenes

## Abstract

Complexes of lanthanide ions, such as Eu(III) (red light emission) and Tb(III) (green light emission), with proper ligands can be highly luminescent and color-tunable, also attaining yellow and orange emission under UV radiation. The ligands employed in this work were poly(sodium acrylate), working as polymeric matrix, and 1,10-phenanthroline, taking advantage of its antenna effect. Possibilities of color display were further enhanced by incorporating a cationic polyfluorene with blue emission. This strategy allowed for obtaining cyan and magenta, besides the aforementioned colors. Uncoated cellulose paper was impregnated with the resulting luminescent inks, observing a strong hypsochromic shift in excitation wavelength upon drying. Hence, while a cheap UV-A lamp sufficed to reveal the polyfluorene’s blue emission, shorter wavelengths were necessary to visualize the emission due to lanthanide ions as well. The capacity to reveal, with UV-C radiation, a full-color image that remains invisible under natural light is undoubtedly useful for anti-counterfeiting applications. Furthermore, both lanthanide ion complexes and polyfluorenes were shown to have their luminescence quenched by Cu(II) ions and nitroarenes, respectively.

## 1. Introduction

Except for La(III) and Lu(III), trivalent lanthanide ions (Ln(III)) have intrinsic luminescent properties, typically with sharp f–f emission lines [1,2]. Nonetheless, two conditions are generally required to yield highly luminescent materials. The first one is the existence of non-centrosymmetric interactions with ligands so that electric dipole transitions, otherwise forbidden by the Laporte rule [3], are partially allowed. Hard Lewis bases, including organic ligands with carboxylate groups [4], are generally favored over soft donors. When these carboxylate groups belong to a polymer such as poly(sodium acrylate) (PSA), this polymer can simultaneously work as a binder, thickener, and support matrix [5]. Still, owing to the low molar extinction coefficients of Ln(III), a second condition is that at least one of those ligands can act as a light harvester, absorbing photons and transferring energy to the metal ion. This is commonly known as antenna effects, and the ligands used for this purpose are usually highly conjugated, such as 1,10-phenanthroline (Phen) [6].

In another context, a different kind of luminescent property is found among conjugated polymers. First, π–π* transitions in highly conjugated systems typically result in strong absorption in the ultraviolet (UV) region, while n–π* transitions are often associated with lower absorption and longer wavelengths, possibly within the visible spectrum [7,8]. In turn, the relaxation of the excited electronic states generally results in fluorescent emission at longer wavelengths [7]. One particular family of conjugated polymers is that of polyfluorenes [9,10].

Both Ln(III)-containing and polyfluorene-containing aqueous systems have been proven useful when coated, painted, or printed onto paper surfaces [11,12]. The advantages of cellulose paper as support or substrate include its renewable origin, its high availability, its relatively low cost and, as long as it is not produced with large amounts of non-biodegradable additives, its disposability [13]. In any case, paper-based materials that support luminescent and optically responsive substances have interesting applications, transcending the traditional usage of paper for display of communication, cleaning, or packaging. Such applications include, for instance, anti-counterfeiting and detection or quantification of pollutants [14,15].

The possibilities of tuning the desired colors in each case readily allow for a straightforward authentication strategy. For instance, Andres et al. [11] attained complex, full-color images to be revealed only under UV radiation, and they did so by using Eu(III) as the source of red emission, Tb(III) for green emission, and an undisclosed fluorescent ink for blue emission. While combinations of asymmetric complexes of Eu(III) and Tb(III) give way to colors such as orange and yellow [16], the incorporation of a luminescent blue ink adds others like cyan and magenta [17].

More possibilities are offered by phenomena of selective luminescence quenching in the presence of specific substances. On the one hand, the emission of Ln(III) complexes is often quenched by electron acceptors that have been postulated to compete with Ln(III) ions for organic ligands [18,19]. However, given the high availability of carboxylate groups, a previous work has discarded the competition for PSA as the quenching mechanism [5]. On the other hand, polyfluorenes undergo fluorescence quenching by mechanisms such as Förster resonance energy transfer or photoinduced electron transfer [20]. Often, a non-fluorescent adduct is formed between the conjugated polymer and aromatic compounds with strong electron-withdrawing groups, such as 2,4,6-trinitrotoluene (TNT) [12].

The interaction between lanthanide ions and anionic polyelectrolytes is a simple and effective strategy to obtain a luminescent lanthanide-containing solid matrix [21,22] due to the loss of the lanthanide hydration shell upon interaction. Following that, this work explores the combinations of Eu(III) and Tb(III) with PSA as a ligand, the relevance of the antenna effect by the incorporation of Phen, and further color tuning under UV radiation by the addition of a polyfluorene. The structures of lanthanide-based complex and the conjugated polymer are schematized in Figure 1. Depending on the purpose, either an organic co-solvent (ethanol) or a surfactant might be used to stabilize Phen in preeminently aqueous media. Then, these aqueous or aqueous/alcoholic systems, constituting luminescent inks by themselves, were coated or painted on paper. Papers coated with Ln(III) complexes suffered quenching when immersed in Cu(II) solutions, while the emission from the polyfluorene was suppressed in the presence of TNT.

## 2. Results

### 2.1. General View on the Usability for Authentication Purposes

The wavelengths at which lanthanide complexes absorb electromagnetic radiation is of utmost importance for their usability. In solution, Ln(III)/PSA/Phen systems displayed high absorptivity between ca. 336 nm and 348 nm, owing to Phen-to-metal energy transfer, along with some characteristic absorption bands that allow us to distinguish Eu(III)-based from Tb(III)-based complexes (Appendix A). Excitation spectra of Eu(III)/PSA and Tb(III)/PSA can be found elsewhere [5].

Indeed, there were significant differences between the required excitation wavelengths for Ln(III)-containing solutions and air-dried papers that had been coated with the same solutions (Appendix A). The same phenomenon was discussed in a previous work of ours [12] with the polyfluorene used in this work: poly(9,9-bis(3′-(*N*,*N*-dimethyl)-*N*-ethylammoinium-propyl-2,7-fluorene)-alt-2,7-(9,9-dioctylfluorene))dibromide, hereinafter referred to as “PFN”. In short, water solvation induced a red shift in absorption.

As indicated in Table 1, coated and dried papers needed to be irradiated at lower wavelengths than their aqueous counterparts. PFN dispersed in water with sodium dodecylsulfate (SDS) at a concentration slightly below the critical micelle concentration [23] displayed fluorescent emission even if subjected to natural light, but it required at least UV-A radiation in the solid state.

In Figure 2, “visible light” refers to halogen white light. The source of UV radiation was an UV-C lamp (254 nm) in Figure 2a,b, and a UV-A lamp (366 nm) in Figure 2c–e. The general purpose of using these inks for authentication is evidenced by the change from visible light to UV radiation in each case. RGB coordinates, easily obtained with a common smartphone and mobile apps such as *Colorimetric Titration* [24], are also displayed in Figure 2b,e, as they could allow for more accurate verification.

In general, wavelengths in the UV-B–UV-C suited both Ln(III) complexes and PFN on paper, as shown by the letters “u” (PSA/Tb(III)/Phen + PFN/SDS) and “s” (PFN/SDS) in the word “inpactus” (Figure 2a). UV-A radiation was insufficient for paper impregnated with Ln(III) complexes, but even an affordable UV-A lamp allowed the user to perceive the fluorescent emission of PFN-coated papers (Figure 2e). It should be noted that the emission of PFN shown in Figure 2e (under UV-A radiation, corresponding to absorption on n–π* transitions) is cyan, but the same polymer gives out blue emission under UV-C radiation (letter “s” in Figure 2a). In the latter case, the cyan color is accomplished by mixtures of PFN with Tb(III), while the magenta color results from mixing it with Eu(III).

### 2.2. Emission Spectra in Solution

Even in the absence of antenna compounds, PSA/Eu(III) and PSA/Tb(III) complexes displayed measurable luminescence in aqueous solution, and so did the mixtures thereof. Emission spectra for an excitation wavelength of 600 nm are shown in Figure 3a. Remarkably, under the same conditions and within the same session of utilization of the spectrofluorometer, the intensity recorded for the main emission band of PSA/Tb(III), 545 nm, was much higher than that of the main emission band of PSA/Eu(III), 616 nm. For low proportions of Eu(III) in the mixtures, the latter peak is even overlapped by the ^5^D_4_ → ^7^F_3_ transition peak of Tb(III) at 622 nm [25].

The evolution in the intensity of Eu(III)’s and Tb(III)’s identifiable peaks, highlighting the electronic transitions associated with them [25,26], are depicted in Figure 3b. It should be noted that this graph omits emission bands for which there is overlapping between Eu(III)’s and Tb(III)’s transitions.

The antenna effect provided by Phen exerted a significant impact on the emission spectra of Eu(III), Tb(III), and their mixtures. The intensity of the emission resulting from the different electronic transitions was unevenly increased. For instance, the characteristic emission bands of Eu(III) were enhanced to a greater extent than those of Tb(III), as can be seen from Figure 4a. Furthermore, emission due to the ^5^D_4_ → ^7^F_6_ transition of Tb(III) was partially overlapped by Phen’s broad band. Hence, this transition is not represented in Figure 4b. The latter graph, besides highlighting the ^5^D_0_ → ^7^F_4_ transition of Eu(III) for comparison with Figure 3b, also shows its ^5^D_0_ → ^7^F_4_ transition, whose prominent emission band was overlapped by one of Tb(III)’s transitions in the case without Phen.

### 2.3. Emission Spectra of Coated Papers

Despite the notorious blue shift in absorption spectra when the solutions of lanthanide ion complexes were coated onto paper, there were no significant differences regarding the emission wavelength of each transition (Figure 5a). Thus, the color of Tb(III)-coated strips and Eu(III)-coated strips under UV radiation matched the solutions used for impregnation. However, there were noticeable differences regarding Phen and the mixtures. In comparison with solutions, the antenna effect favored to a greater extent Eu(III) over Tb(III). For instance, in the 50% case (1:1), the ratio of the emission intensity of europium’s ^5^D_0_ → ^7^F_2_ transition (616 nm) to that of terbium’s ^5^D_4_ → ^7^F_5_ transition (545 nm) was 4.04 for air-dried coated papers. The same ratio was 2.22 in the case of Phen-containing solutions and 0.49 in the case of solutions without Phen. Indeed, as displayed in Figure 5b, the curvature of the function of the emission intensity assigned to each transition with the molar ratio of Tb(III) changed from that of Figure 4b (solutions).

From the integrated ^5^D_0_ → ^7^F_2_ emission peak and the excitation spectrum monitored at 616 nm, the quantum yield of Eu(III)/PSA/Phen-coated paper was estimated as 45%. That of Tb(III)/PSA/Phen-coated paper, on the basis of its ^5^D_4_ → ^7^F_5_ emission peak (545 nm), was 21%. These values are similar to those found for Ln(III)/PSA/Phen composites in the solid state [16].

### 2.4. Further Color Tuning with Polyfluorenes

A challenge to overcome when incorporating PFNBr into the same matrix as the aforementioned Ln(III)/PSA/Phen complexes was the poor solubility of the cationic polyfluorene in water/ethanol mixtures. In other words, precipitation of PFNBr was observed unless ethanol was replaced with a proper co-solvent or surfactant [27]. After some failed preliminary tests with methanol, acetonitrile, and non-ionic surfactants, we opted for an anionic surfactant, SDS, to stabilize an aqueous system.

Figure 6a shows the emission spectra of these systems without ethanol or any other organic solvent, i.e., PSA/SDS/Phen with Eu(III), Tb(III), and PFN, and their combinations: Eu(III) with Tb(III) (1:1, mol), Eu(III) and PFN (9:1, mol), Tb(III) and PFN (9:1, mol), and Eu(III) with both Tb(III) and PFN (9:1:1, mol). For PFN, the amount of substance in mol was calculated on the basis of its repetitive unit.

Combining Eu(III) with PFN allowed a magenta ink to be obtained under UV radiation (Figure 6b), while Tb(III) and PFN resulted in cyan (Figure 6c). These colors were not possible to attain with only Eu(III) and Tb(III). Finally, somehow due to the addition of its multiple emission bands, the mixture of PFN and the lanthanide ions gave out a purple color under UV radiation, corresponding to the first vial in Figure 6d. These inks kept their luminescence when painted on paper (Figure 6e) but required excitation at lower wavelengths.

### 2.5. Quenching of Luminescence

It is well-known that certain metal ions quench the luminescence of lanthanide ions in polymeric matrices. Studies with Eu(III)/PSA and Tb(III)/PSA have already been reported [5], and Cu(II) was the cation accounting for the highest suppression of luminescence in both cases.

Likewise, the fluorescence of polyfluorenes and other conjugated polymers is quenched by strong electron-accepting compounds, such as nitroaromatics. More precisely, the emission of the polyfluorene involved in this work is non-selectively reduced by 1,2-dinitrobenzene, 2,4-dinitrotoluene and TNT [12,28].

As displayed in Figure 7a, the luminescence of paper strips coated with Ln(III)/PSA/Phen was quenched in an identifiable and quantifiable way by immersion in copper(II) nitrate solutions, at least in the millimolar range. The quenching efficiency (*QE*) was computed from emission intensities in the absence and presence of quencher (*I*_0_ and *I*, respectively) and recordings during the same session:(1)QE(%)=100(I0−I)I0

The fitting lines in Figure 7a correspond to hyperbolic functions that are discussed below (viz. Section 3).

## 3. Discussion

### 3.1. Insights into the Antenna Effect and Wet/Dry Differences

When both Tb(III) and Eu(III) were present, the antenna effect apparently favored the latter. Without Phen and with an Eu(III)/Tb(III) ratio of 1:4, the ratio of the emission intensity of Tb(III)’s most prominent band to that of Eu(III) was 5.98. With Phen, it was 1.56. This can be partially explained by quantitative differences in energy transfer from Phen to the metal center, as the intensity at ca. 348 nm in excitation spectra (Appendix A) was higher for Eu(III)/PSA/Phen than for Tb(III)/PSA/Phen.

In addition, their 1:1 mixture displayed an excitation spectrum that, if monitored at 616 nm, clearly resembled the spectrum of Eu(III)/PSA/Phen. While the intensity of every characteristic band was lower, the ^7^F_0_ → ^5^L_6_ electronic transition in europium produced a prominent peak in the case of the 1:1 mixture at ca. 395 nm [29,30]. In contrast, the characteristic peak of Tb(III)’s ^7^F_6_ → ^5^L_10_ electronic transition, found at 369 nm [31], could not be identified in excitation spectra monitored at 616 nm. Inversely, when the monitoring intensity was 545 nm, no characteristic peaks of Eu(III) were found. This may corroborate what was postulated in a previous work [16]: (i) there is preferential Phen-to-Eu(III) energy transfer; (ii) there is certain non-reversible Tb(III)-to-Eu(III) energy transfer. Indeed, the latter effect is limited by the presence of ligands between lanthanide ions, as the strength of energy transfer phenomena is strongly reduced with increasing interatomic distance [32]. However, as depicted in Figure 1, they may not be completely coordinated with PSA and Phen. In solution ([Eu^3+^] = 10 mM, [Tb^3+^] = 50 mM), the Tb(III)-to-Eu(III) energy transfer efficiency in systems involving PSA and Phen has been estimated as ca. 46% [16]. This energy transfer phenomenon implied that the decay lifetime of Tb(III) decreased with increasing Eu(III) concentration [33].

In other works in which energy transfer was essential for the anticounterfeiting system, this transfer from certain sensitizer ions, such as Ce^3+^ or Yb^3+^ [34], to other lanthanide ions was promoted in core-shell nanostructures, e.g., by means of Gd^3+^ [35,36]. However, in the system presented here, the role of Tb(III) as sensitizer to enhance Eu(III)’s emission was detrimental to the purpose of color tuning. However, in comparison with tunable solutions [16], less proportion of Tb(III) was necessary to obtain orange and yellow colors on paper. The post hoc hypothesis presented to explain this is that the loss of water molecules by evaporation shifted the complex formation equilibria towards a case of total coordination with PSA and Phen. Overall, virtually all possibilities of complex formation are considered in the following stoichiometric equation:(2)$${\mathrm{LnCl}}_{3}\cdot 6H_{2}O+x\mathrm{COONa}+y\mathrm{Phen}+\left(3-x\right)\mathrm{NaOH}\overset{}{\Leftrightarrow }{\left[\mathrm{Ln}{\left(H_{2}O\right)}_{6-2x-2y}{\left(\mathrm{Phen}\right)}_{y}{\mathrm{COO}}_{x}\right]}^{3-x}+3\mathrm{NaCl}+\left(2x+2y\right)H_{2}O+\left(3-x\right){\mathrm{OH}}^{-}$$

A case of total coordination implies that *x* = 2 and *y* = 1 since there was no excess of Phen. The complex schematized in Figure 1 considers that *x* = 1 and *y* = 1. Of course, fractional values of *x* and *y*, involving different extents of coordination, are also plausible. What the post hoc hypothesis means is that, as water is evaporated, Equation (4) adopts increasingly higher values of *x* and *y*. Once dry, complexes are highly coordinated with PSA and Phen, hindering energy transfer between lanthanide ions and requiring shorter wavelengths for excitation.

### 3.2. On Luminescence Quenching

Even in the absence of quencher ions (i.e., Cu^2+^), immersion in water reduced the effective photoluminescence of Ln(III)-coated papers, most likely by simple elution. If this effect by elution (*QE*_0_) is considered to be simply additive to the *QE* by Cu(II), a satisfactory fitting to a hyperbola is attained both for Eu(III) (*R*^2^ = 0.99) and Tb(III) (*R*^2^ = 0.97), as shown in Figure 7a. The expression of this fitting equation follows:(3)$$\left[QE\left(\%\right)-QE_{0}\left(\%\right)\right]=\frac{100\;k\;\left[\mathrm{Cu}\left(\mathrm{II}\right)\right]}{1+k\;\left[\mathrm{Cu}\left(\mathrm{II}\right)\right]}$$

It can be seen that in Equation (3), the constant *k* corresponds to the Stern–Volmer constant (*K*_SV_) in a linear plot against *(I*_0_*/I)*’, which is the corrected ratio of the intensity without a quencher to the intensity with quencher. This correction implies the suppression of the elution effect, i.e., it assumes the blank sample (strip soaked in water) to be the system without a quencher. Thus, the Stern–Volmer plot can be expressed as [37]:(4)(I0I)′=1+KSV [Cu(II)]

Then, *K*_SV_ adopts a value of 0.36 mM^–1^ for Eu(III) and a value of 0.20 mM^–1^ for Tb(III) in coated paper. Nonetheless, complete quenching was possible in both cases if the concentration of the Cu(II) solution was high enough. Therefore, considering the standard deviations of the blank and the other measurements, it can be indicated that the proper range of quantification is from 1 mM to 10 mM, although the limit of detection is roughly 0.5 mM. These parameters point out a worse analytical performance than that of the lanthanide ion complexes in solution [5,38]. Nonetheless, in consideration of the mechanism discussed above, reducing their concentration in the inks may allow for detecting lower concentrations of Cu(II). That said, a thoughtful optimization of the luminescent inks presented herein for analytical purposes lies out of the scope of this work. Such purposes are addressed elsewhere in the literature. For instance, metal-organic frameworks with Eu(III) and triazole sites attained a limit of detection for Fe^3+^ of only 0.018 mM [39].

In any case, the potential offered by selective luminescence quenching is not limited to analytical purposes. When it comes to authentication, being quenched by Cu(II) ions is an additional feature of the material, and this unique characteristic might not be easy to counterfeit.

Regarding the quenching of PFN by TNT, it can be noted that lower concentrations of quencher were required to observe a significant influence. This is explained by the “amplified quenching effect” of conjugated polymers, where a single quencher molecule can reduce or suppress the fluorescence emission of a whole polymer chain [40]. It is generally accepted that the mechanism by which it does so is electron transfer [41]. Nitroarenes such as TNT are strong electron acceptors, tending to participate in charge transfer phenomena with nucleophiles and with virtually any electron-rich molecule or moiety. This includes the backbone of polyfluorenes, whose delocalized π electrons define the optical properties of the macromolecule [40]. The loss of one electron from one of the fluorene units can result in noticeable quenching along the entire conjugated structure.

## 4. Materials and Methods

### 4.1. Materials

PSA (average molecular weight 2100 g mol^–1^), Phen, EuCl_3_·6H_2_O, TbCl_3_·6H_2_O, Cu(NO_3_)_2_, 2,4-dinitrotoluene (DNT), sodium dodecyl sulfate (SDS), sodium hydroxide, and PFN were received from Sigma-Aldrich (branch office in Lisbon, Portugal). Organic solvents, including acetone, absolute ethanol, acetonitrile, and methanol, were purchased from Thermo Fisher Scientific (Porto Salvo, Portugal). TNT was synthesized from DNT as described elsewhere [12]. All references to water in this work correspond to Milli-Q grade water.

The substrate to be impregnated was uncoated paper sheets of industrial origin with basis weight 90 g m^–2^, tensile index 95 ± 2 N m g^–1^, Gurley air resistance 18 ± 1 s/100 mL, and Bendtsen roughness 86 ± 10 mL/min.

### 4.2. Preparation of Inks

Stock solutions of TbCl_3_·6H_2_O/PSA, EuCl_3_·6H_2_O/PSA, TbCl_3_·6H_2_O/PSA/Phen, and EuCl_3_·6H_2_O/PSA/Phen were prepared in ethanol/water (1/4, *v*/*v*) solvent systems, in such a way that the concentration of each component equaled 50 mM. On the one hand, the first two stock solutions were physically mixed at eleven different volume ratios and had to be diluted to half for analysis (fluorimetry) in order to avoid attaining saturation intensity in the emission spectra. On the other hand, TbCl_3_·6H_2_O/PSA/Phen and EuCl_3_·6H_2_O/PSA/Phen (50 mM) stock solutions were also mixed at different ratios. The strongly luminescent inks that resulted from this mixture, describing a bright red–orange–yellow–green range under UV radiation, were directly used for paper impregnation with the purpose of clear visualization. Nonetheless, they had to be diluted to 10 mM to avoid saturation when recording emission spectra.

Three additional stock solutions were prepared in strictly aqueous systems, replacing ethanol with SDS. These solutions encompassed TbCl_3_·6H_2_O (50 mM), EuCl_3_·6H_2_O (50 mM), and PFN (1 mM, on the basis of the monomer), consistently in a PSA (50 mM)/Phen (50 mM)/SDS (8 mM) aqueous matrix. They were directly used for paper impregnation or diluted five times with the aforementioned aqueous matrix for spectrofluorometry.

In all cases, pH was adjusted to 7 with small additions of NaOH to ensure the quantitative ionization of PSA.

### 4.3. Dip Coating

The uncoated paper was cut into rectangular strips that were 15 mm wide and 200 mm long. Impregnation of paper with luminescent inks was performed by means of a KSV Nima Dip Coater (Biolin Scientific AB, Västra Frölunda, Sweden), leaving each strip immersed for 1 min. Dip-coated paper strips were then dried for 5 min by a thermoventilator at approximately 50 °C. Alternatively, some inks were directly applied onto paper by hand, using a paintbrush, and left to dry at 23 °C, with a relative humidity of approximately 50% for 24 h.

### 4.4. Spectrofluorometry

Preliminarily, excitation spectra of all solutions and coated papers were recorded by means of a Jovin-Yvon Spex Fluorog 3-2.2 spectrofluorometer (Jobin Yvon Inc.—Horiba, NJ, USA) in order to obtain a range of plausible excitation wavelengths in each case. Then, emission spectra of solutions were collected on this spectrofluorometer, set with the following configuration: right angle, slit width of 1 nm, and an integration time of 1 s. The device included a 450 W ozone-free xenon arc lamp as a light source [42]. The excitation wavelength was 366 nm for aqueous/alcoholic inks and 348 nm for SDS-stabilized aqueous systems.

The same device was used to record the spectra of paper samples but opted for a front-face (45°) configuration, a slit width of 0.5 nm, and an integration time of 1 s. The excitation wavelength was 320 nm.

### 4.5. Quenching Assays

Cu(NO_3_)_2_ solutions of different concentrations (1 mM–15 mM) were prepared in water. TNT solutions with concentrations ranging from 0.1 mM to 2.5 mM were prepared in acetone. Paper strips coated with Ln(III) complexes were immersed for 1 min in Cu(NO_3_)_2_ solutions, while those strips with PFN were immersed in the TNT solution, also for 1 min. Before spectrofluorometry and photographing the strips, they were left to dry in an exicator for 24 h. Likewise, they were protected from light with aluminum foil until characterized. Emission spectra were recorded during the same session in each case, following the procedure described above for coated papers in general.

## 5. Conclusions

Combinations of Eu(III) and Tb(III) ions with each other, with polymeric ligands (PSA), with antenna co-ligands (Phen), with co-solvents (ethanol) or surfactants (SDS), and with a cationic polyfluorene shed some light into their energy transfer interactions, applications, and limitations. Here, a list of highlights from the results follows:Mixtures of Tb(III)/PSA and Eu(III)/PSA resulted in luminescent solutions with tunable color in the green-to-red range, including yellow and orange, but with low emission intensity.The antenna effect, essentially consisting of energy transfer from Phen to the metal center of the complex (either europium or terbium), greatly increased luminescence but favored Eu(III) over Tb(III).Adding a polyfluorene, stabilized in an aqueous medium with SDS, and avoiding the use of co-solvents that would result in the precipitation of PFN, allowed us to attain a blue–red color range, including cyan and magenta.Impregnated papers, once dry, required shorter wavelengths of UV radiation for proper excitation than their aqueous counterparts.Color tuning possibilities, by themselves, allowed for anti-counterfeiting applications, as papers that look white under natural light can display complex full-color information under UV radiation.Although Cu(II) was proven an effective quencher for Ln(III)-coated papers, attempts to implement this feature in paper-based analytical devices faces the limitation of a high limit of detection. Instead, responsiveness towards Cu(II) can also be taken advantage of for anti-counterfeiting purposes.

## Figures and Tables

**Figure 1 molecules-28-06164-f001:**
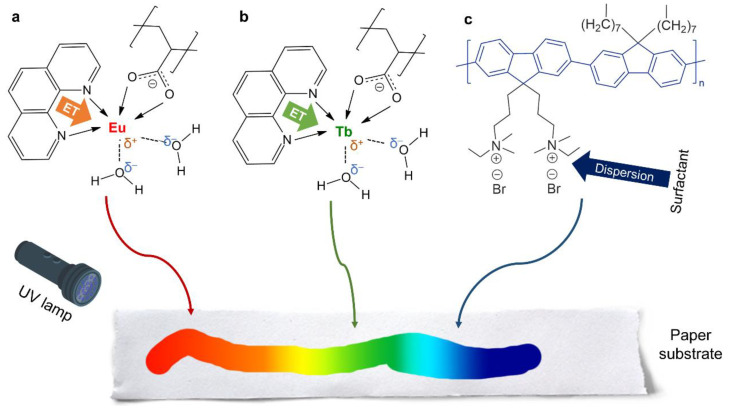
Plausible complexation of (**a**) Eu(III) and (**b**) Tb(III) with both PSA and Phen, and (**c**) the polyfluorene involved in this work. ET: energy transfer. The complete coordination of the hexadentate ions (e.g., implying two carboxylate groups from PSA) should not be ruled out.

**Figure 2 molecules-28-06164-f002:**
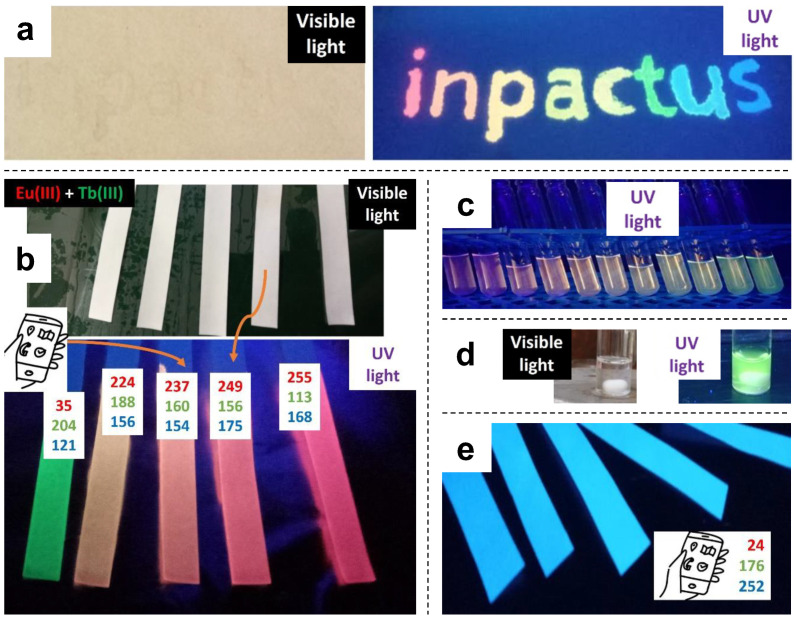
Proof of concept for anti-counterfeiting applications: (**a**) paper sheet written with luminescent inks, both under natural light and under UV radiation; (**b**) paper strips coated with lanthanide-based complexes; (**c**) solutions of lanthanide-based complexes for spectrofluorometry; (**d**) Tb(III) complex solution under visible and UV light; (**e**) paper strips coated with PFN.

**Figure 3 molecules-28-06164-f003:**
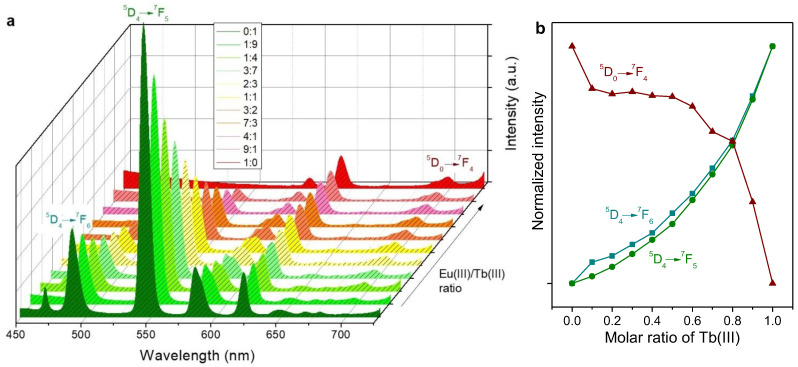
Solutions of Eu(III)/PSA, Tb(III)/PSA, and mixtures thereof: (**a**) emission spectra under excitation at 366 ± 0.5 nm; (**b**) trends in the emission intensity corresponding to some characteristic transitions. The legend indicates the molar ratio (Eu:Tb) in each case.

**Figure 4 molecules-28-06164-f004:**
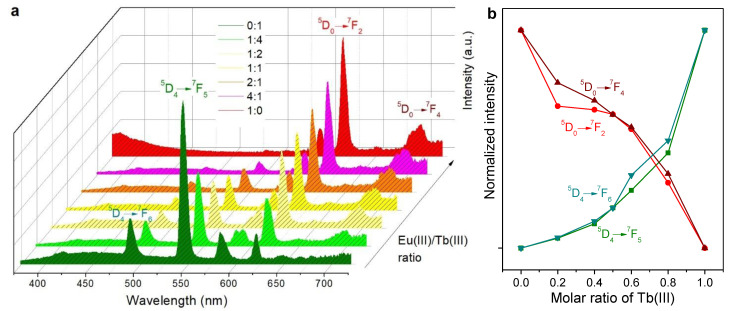
Solutions of Eu(III)/PSA/Phen, Tb(III)/PSA/Phen, and mixtures thereof: (**a**) emission spectra under excitation at 366.0 ± 0.5 nm; (**b**) trends in the emission intensity corresponding to some characteristic transitions. The legend indicates the molar ratio (Eu:Tb) in each case.

**Figure 5 molecules-28-06164-f005:**
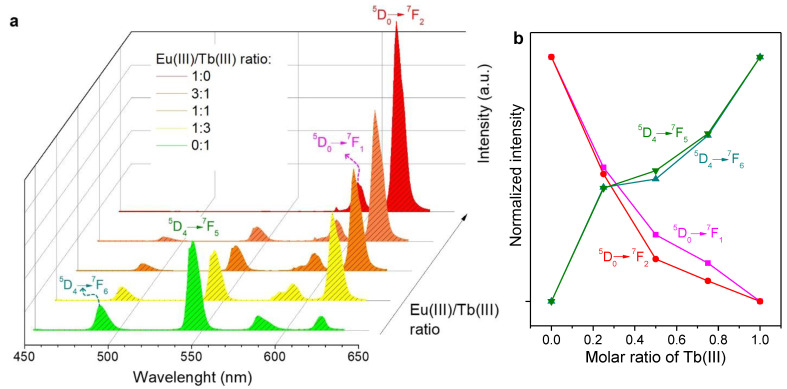
Paper strips coated with Eu(III)/PSA/Phen, Tb(III)/PSA/Phen, and mixtures thereof: (**a**) emission spectra under excitation at 320.00 ± 0.25 nm; (**b**) trends in the emission intensity corresponding to characteristic transitions. The legend indicates the molar ratio (Eu:Tb).

**Figure 6 molecules-28-06164-f006:**
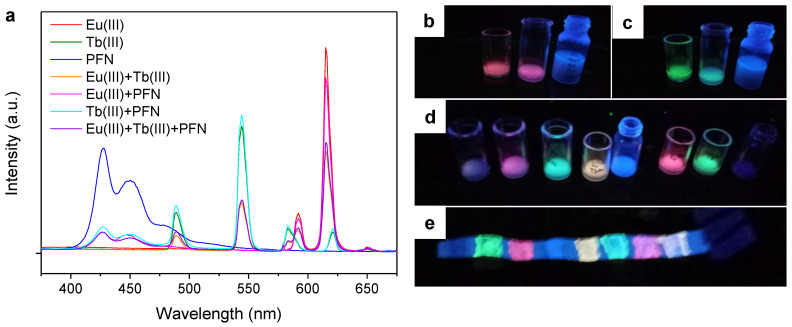
PSA/SDS/Phen matrix with different proportions of Tb(III) ions, Eu(III) ions, and PFN: (**a**) emission spectra under excitation at 348 nm, (**b**–**d**) pictures of the solutions under radiation at 365 nm, and (**e**) picture of their use on a paper strip under radiation at 254 nm.

**Figure 7 molecules-28-06164-f007:**
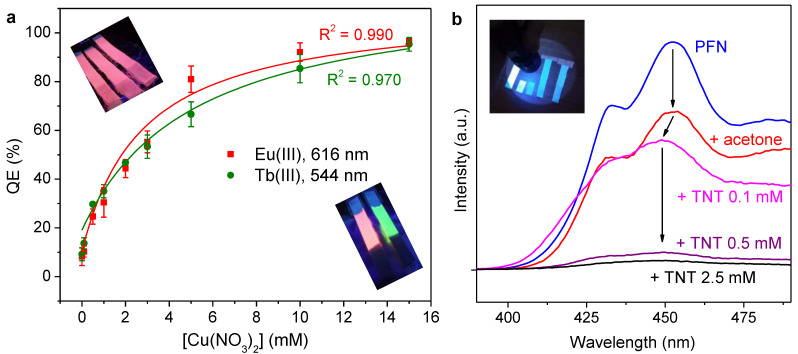
Luminescence quenching: (**a**) quenching efficiency of paper strips with lanthanide ions by the effect of Cu(II), including inset photographs that illustrate the case of partial quenching (up) and total quenching (bottom); (**b**) emission spectra of PFN-coated paper strips without and with immersion in TNT/acetone solutions. The inset photograph in (**b**) was taken with a commercially available UV-A lamp (395 nm).

**Table 1 molecules-28-06164-t001:** Appropriate choice of excitation wavelength when using the luminescent ink systems proposed in this work, either in solution or dried on cellulose paper.

System	In Solution	On Paper
Ln(III)/PSA	Eu(III): UV-A; Tb(III): UV-B/UV-A	UV-B/UV-C
Ln(III)/PSA/Phen	UV-A/UV-B. Max. ~ 336–348 nm	UV-B/UV-C. Max. < 270 nm
PFN/SDS	Visible–UV-B. Max. ~ 413 nm	UV-A–UV-C. Max. ~ 370 nm

## Data Availability

The CSV files corresponding to emission and absorption spectra will be made publicly available in the largest Portuguese data repository, https://www.rcaap.pt/ (accessed on 1 July 2023).

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
