# Peer review of "Luminescent Papers with Asymmetric Complexes of Eu(III) and Tb(III) in Polymeric Matrices and Suggested Combinations for Color Tuning"

_molecules, 2023, doi:10.3390/molecules28166164_

Round 1

Reviewer 1 Report

In this manuscript, the lanthanide-based complexes (e. g. Eu(III) and Tb(III)) and the conjugated polymers have been synthesized and the luminescent properties of the as-obtained complexes are also investigated. Interestingly, the multicolor emissions could be realized via the change of Eu(III)/Tb(III) ratio. As an example, the application of the as-obtained samples as an anti-counterfeiting agent and fluorescent probe is described. I think this manuscript can be accepted for publication after the authors modify the following issues.

(1). The energy transfer (ET) processes from PSA/Phen to lanthanide ions (Eu/Tb (III)) and Tb to Eu exist in the system. The ET processes are of significance for the luminescence. The ET efficiency should be provided in this manuscript?

(2) The decay lifetimes of Eu and Tb ions should be given in this manuscript.

(3) The luminescent quantum yield (QY) of lanthanide-based complexes is an important parameter. Thus, the QYs of lanthanide-based complexes could be provided in this article.

(4) More related references can be read and cited in this manuscript.

1. Advanced Materials, 2020, 32(45): 2002121.

2. Nanotechnology, 2020, 31(36): 365705.

3. Journal of Materials Chemistry C, 2016,4, 2432-2437.

The writing has a few grammatical and stylistic errors, and these should be fixed prior to publication.

Reviewer 2 Report

Aguado et al. investigated the luminescent properties of lanthanide ion complexes, specifically Eu(III) emitting red light and Tb(III) emitting green light, with the incorporation of appropriate ligands. By utilizing poly(sodium acrylate) and 1,10-phenanthroline ligands, they harnessed the antenna effect for enhanced performance. Additionally, they introduced a cationic polyfluorene with blue emission to expand the color range, achieving cyan and magenta emissions in addition to the existing colors. Practical application was demonstrated by impregnating luminescent inks onto uncoated cellulose paper, revealing a significant shift in excitation wavelength upon drying. The authors also explored the potential of anti-counterfeiting applications, utilizing UV-C radiation to unveil a hidden full-color image. Notably, they observed luminescence quenching of both lanthanide ion complexes and polyfluorenes when exposed to Cu(II) ions and nitroarenes, respectively. The study contributes valuable insights into the intricate luminescence behavior and material interactions, with implications for anti-counterfeiting and potential practical uses.

Comments:

1. Mention the slit width in all the Figure captions for the emission study.

2. Assign the emission peaks Eu(III) and Tb(III) in Figure 5, 6, and & 7.

Reviewer 3 Report

1. Why not check the properties of the mixture Tb/Eu-complex in one system? Different ration?

2. How could you check the right coordinated mode? Pls do their characterization

3. From the Fig.7, I suggest the authors fit the quenching equation and discuss its quenching behavior.

4. Some refs on the Tb/Eu-complex on the PL feature may be considered, such as J. Mol. Struct. 1282 (2023) 135220; ChemPlusChem, 2016, 81:1299-1304.
